# Women's empowerment and contraceptive use: Recent evidence from ASEAN countries

**Ferry Efendi**[1]*, **Susy Katikana Sebayang**[2], **Erni Astutik**[2], **Sonia Reisenhofer**[3], **Lisa McKenna**[3]

1 Faculty of Nursing, Universitas Airlangga, Surabaya, Indonesia, 2 Department of Epidemiology, Biostatistics, Population Studies and Health Promotion, Faculty of Public Health, Universitas Airlangga, Surabaya, Indonesia, 3 School of Nursing and Midwifery, La Trobe University, Melbourne, Australia

* ferry-e@fkp.unair.ac.id

## Abstract

### Background

A fundamental element of gender equity are women's rights to reproductive choice. Women's empowerment is often linked to enabling decisions around contraceptive use and reduced fertility worldwide, although limited evidence is currently available around contraceptive use and decision making in ASEAN countries.

### Objective

To examine the association between women's empowerment and contraceptive use in five selected ASEAN member states.

### Methods

Data from the latest Demographic and Health Survey of Cambodia, Indonesia, Myanmar, The Philippines, and Timor-Leste were used. The main outcome was contraceptive use among married women (15–49 years) from these five countries. We considered four indicators of empowerment: labor force participation; disagreement with reasons for wife beating; decision-making power over household issues; and knowledge level.

### Results

Labor force participation was found to be significantly associated with contraceptive use in all nations. Disagreement with justification of wife beating was not significantly related to contraceptive use in any country. Decision-making power (higher) was only associated with contraceptive use in Cambodia, while higher knowledge levels were associated with contraceptive use in Cambodia, and Myanmar.

### Conclusion

This study suggests women's labor force participation is an important determinant of contraceptive use. Policies designed to open the labor market and empower women through education should be implemented to enable women's participation. Gender inequality may

**Data Availability Statement:** Data is available from the https://dhsprogram.com/data/new-user-registration.cfm by submitting an application to the ICF via the website. Other researchers will be able

to access the data set in the same way as the authors, and the authors do not have special access rights that others do not have. Interested researchers can replicate the findings in this study as a whole by directly obtaining data from IDHS by following the protocol in the manuscript method section.

**Funding:** F.E. received funding support from the Ministry of Research, Technology and Higher Education of Indonesia during his post-doctoral degree. All the funding sources of support (whether external or internal to your organization) was not involved in study design, data collection, analysis or interpretation; in the writing of this report; or in the decision to submit the article for publication. There was no additional external funding received for this study.

**Competing interests:** We have declared that we don't have any conflict of interest.

also be tackled by engaging women in decision-making processes at national, community and family levels.

## Introduction

Women's empowerment makes a significant contribution to improving overall maternal, neonatal and child health outcomes [1], including within the member countries of the Association of Southeast Asian Nations (ASEAN) [2]. In a recent report on the Millenium Development Goals (MDGs) achievement and preparation toward the Sustainable Development Goals (SDGs), only a few countries achieved the target child mortality reduction or maternal mortality reduction [3]. Despite these countries' successful experiences, many ASEAN nations are still working to achieve MDG targets and move forward to SDG achievement [4]. The premise of the SDGs is to "leave no one behind," with goal five focusing on the empowerment of women and on gender equality.

Women's empowerment according to Kabeer (1999) is the ability of women to make life choices, combining three interrelated dimensions: resources (access, material and human and social); agency (decision-making processes, less measurable manifestations of agency such as negotiation, deception, and manipulation); and achievement (welfare outcomes) [5]. Women's empowerment is a complex issue [1, 6]. Research shows a positive relationship between women's empowerment and reproductive health [7]. In one aspect, women's empowerment is associated with use of female contraceptives in low- and middle-income countries [8]. Studies suggest that understanding the causal impact of women's agency on contraceptive use will help inform policies and programs to increase women's contraceptive choice [9]. Previous studies have shown that women who are more empowered are more likely to have access to antenatal care [10, 11], provide appropriate nutrition to children [12], and use contraception [9, 13]. Empowered women are better able to determine their own contraceptive use [14], delay childbirth if they wish, and negotiate family size [15]. Being able to make these choices is an essential element of women's human rights and decreasing gender inequality [16] and patriarchal values in Asian countries [17].

Family planning is used in some ASEAN member states to reduce fertility in order to slow the rate of population growth. In 2016, the total fertility rates in Cambodia, Indonesia, Myanmar, and The Philippines ranged from 2.21 to 2.92, while in Timor-Leste it was 5.50 [18]. Although there are declining trends in some countries, improving access to family planning and contraceptive services remains fundamental to managing population growth and providing women with choice [19, 20]. As the ASEAN member states work towards the SDG of providing universal access to reproductive health, family planning and access to contraception, it is important to understand factors that promote women's abilities to manage their reproductive health. Our current understanding of the relationship between women's empowerment and contraception use remains incomplete. Effectively measuring empowerment is challenging due to the complexity of the concept [21]. However, it has been achieved using the multi-faceted empowerment module of the Demographic and Health Survey (DHS) [9, 10, 22].

The issue of women's empowerment is central as ASEAN member states work to improve the gender inequality index. This is particularly relevant to Cambodia, Indonesia, Myanmar, and The Philippines, consecutively ranked as 116, 114, 106, and 97 on the gender inequality index in 2017 [23]. The agenda to empower every woman can be a means of overcoming barriers to contraceptive use [24]. In ASEAN states, women have increasing access to family

planning within a fundamental human right to securing good health and women's autonomy [25]. Recent data on family planning and contraceptive access among ASEAN member states showed the largest growth over the last decade, although disparities between ASEAN countries exist [26]. It is thought that women's empowerment plays an important role in women's abilities to use contraception.

In this analysis, we investigated the association between contraceptive use and women's empowerment in five ASEAN countries using data from country specific DHS surveys. The role of women's empowerment is measured by resource components which includes labor force participation and agency component which comprises disagreement with justification for wife beating, decision-making power over household issues, and knowledge level [5, 22, 27–29]. These three interrelated dimensions, resources, agency and outcomes require individual choice to choose among alternatives [5], thus giving women access to education, occupation, and participation in family and environment is inevitable for empowering them [5, 27]. Contraceptive prevalence rates among married women in Cambodia, Indonesia, Myanmar, and The Philippines ranged from 52%-62%, higher than Timor Leste where contraceptive use was only 26% [30–34]. To facilitate the design of regional initiatives and campaigns that will promote gender equality and promote women's abilities to make decisions around contraceptive use, we investigated the association between women's empowerment and contraceptive use in these five ASEAN member states.

## Materials and methods

### Data

Data for the study were derived from recent surveys of DHS for five countries within ASEAN, namely Indonesia (2017), Cambodia (2014), The Philippines (2017), Myanmar (2016), and Timor-Leste (2016). DHS provides a comprehensive picture of population, family planning, reproductive health, and maternal and child health conditions in included countries. Women of reproductive age (15–49 years) in sampled households were eligible for individual interviews. DHS data represent response rates of 98% of women for Indonesia, Cambodia, and The Philippines, 97% of women for Timor Leste and 96% of women for Myanmar. The sample for this study consisted of married women aged 15 to 49 years old from the five years preceding the most recent survey in each country. We excluded never-married and previously married women from this study, as well as participants with missing information about current contraceptive use or missing covariate data. For all analyses, we employed complete cases. The original sample size that met the requirements in Cambodia was 11,898, and after exclusions, the number of samples available for analysis was 10,403. In Indonesia, the total sample size for analysis was 31,495, based on original sample size of 35,681. There were 6,047 samples available for analysis out of 7,759 original samples in Myanmar. The original sample size in the Philippines was 15,016 people, and the sample size available for analysis was 12,815. In Timor-Leste, there were 5,011 samples for analysis out of 7,697 original samples.

DHS is an international survey conducted in over 90 countries and the data are publicly available from http://www.dhsprogram.com/data/available-datasets.cfm. DHS utilizes a complex sampling design, stratifying participants by region and urban/rural status prior to sampling households. Full details of sampling techniques used for obtaining the information have been published at the DHS website (ICF 2018). An Institutional Review Board of the Inner City Fund (ICF) Macro Institutional Review Board approved the proposal by ICF and each implementing country approved the survey protocols. All participants provided verbal informed consent prior to data collection (The DHS Program-ICF 2013). This current study

uses secondary data analysis of a de-identified dataset and additional consent was not required, although approval to use the data for this analysis was granted by ICF International.

## Variables and measurement

In this study, the outcome variable was contraceptive use which was derived from women's responses when they were asked "Are you or your partner currently doing something or using any method to delay or avoid getting pregnant?" in each country. The outcome variable had two categories; currently using and not using contraceptive. Women were classified as "currently using contraceptive" if the response was "yes" and "currently not using contraceptive" if the response was "no". Women who reported being pregnant at interview were coded as not currently using any method.

We adapted the framework of women's empowerment based on previous studies conducted by Phan (2015) and Sebayang, Efendi, and Astutik (2019) [11, 22]. Our framework included women's empowerment in the areas of participation in the labor force, disagreement to justification toward wife beating, decision-making power over household issues, and knowledge level (Fig 1).

We used Principal Component Analysis (PCA) to create empowerment components. PCA enabled creating women empowerment proxies in order to investigate the independent effects

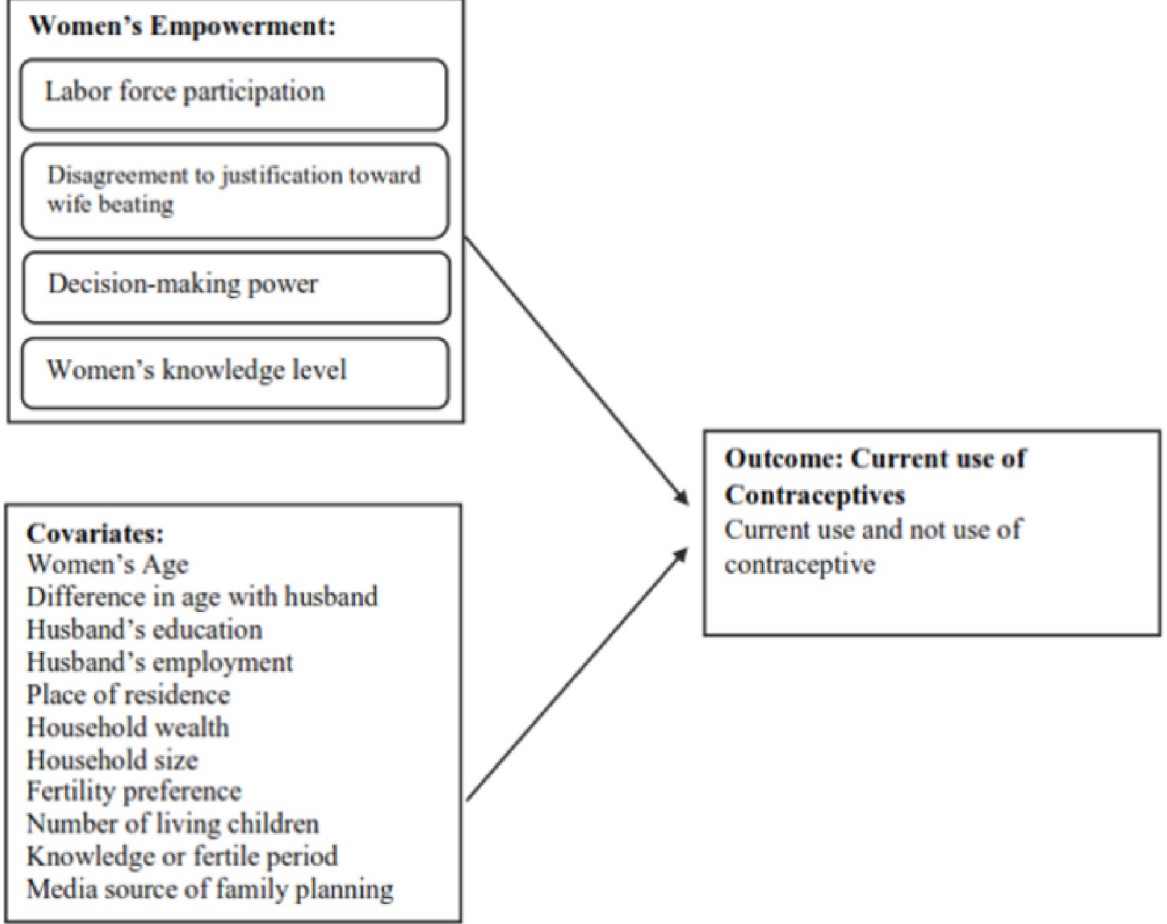

**Fig 1. Conceptual framework of the association between women's empowerment and current use of contraceptives.**

of women's empowerment on certain health outcome [28]. These four components of women's empowerment were calculated from 17 indicators from the DHS surveys.

Labor force participation was indicated by six aspects: work in the last 12 months (yes or no), for whom the woman worked (not working, work for family members, work for someone else, self-employed), women's occupation type (not working, unskilled labor, skilled labor, professional, and self-employed), types of payment (not working or working but not paid, working and paid in cash and in-kind, and working and paid only in cash), working all years (not working, work occasionally, work seasonally, work all year), and earning more than husband (not working or not paid in cash, worked and earned less than husband, worked and earned about the same as the husband, worked and earned more than husband). The second component of attitudes toward wife beating included five aspects: neglecting children, going places without husband's permission, arguing, refusing sex and burning food. Married women between the ages of 15 and 49 years who disagree with the five reasons for wife beating are considered empowered in DHS. If married women agree/confirm at least one of the above-mentioned reasons for wife beating, they are considered helpless [35].

Women's involvement in household decision-making included questions about who made decisions about women's own health care, household purchasing, visiting family, and husband's earnings. All decision components were classified as made by husband or other, made jointly with the woman, or made solely by the woman. The highest score is given to women who can make decisions themselves, the lower score is given to women who make decisions together, and the lowest score is given to women who have no involvement in decision-making.

The last component on women's knowledge includes access to media (newspaper, radio, and television) and the highest level of formal education. Access to media was divided to three categories, namely not at all, some media, all media. Highest level of formal educational was categorized into no education, primary, secondary and higher education.

The seventeen variables of women's empowerment were subjected to principal component analysis, which yielded four distinct components representing labor force participation, disagreement with justification for wife beating, decision-making power, and knowledge level. Each component score was then classified into three tertiles to create categories of low, medium, and high. The full report of methodological aspects can be found elsewhere [11]. High disagreement meant the women disagreed with the highest third of all the justifications for wife beating, medium disagreement meant they disagreed with the middle third of the number of justifications for wife beating, and low disagreement meant they disagreed with the lowest third of the number of justifications for wife beating.

Other covariates known to be associated with contraceptive use were added to the analysis [36]. These included age (categorized into five-year groups, namely 15–19, 20–24, 25–29, 30–34, 35–39, 40–44, 45–49 years), husband's education (incomplete primary or none, complete primary or some secondary, completed secondary or higher), place of residence (rural or urban), household members (>4 or ≤4), socioeconomic status (very poor, poor, middle income, rich, and very rich), husband's occupation (agricultural, non-agricultural), age difference between husband and wife (women older than man, husband 0–4 years older, husband 5–7 years older, and husband >7 years older), fertility intention (wants more children, wants no more children, and undecided), number of living children (none, 1–2, 3–4, and 5+), knowledge of fertile period (correct or incorrect), and in the past few months, having heard about family planning via the radio, TV, or newspaper.

Data analysis took into account the complex survey sampling design which produced the weighted results. Respondents' characteristics, the component of women's empowerment, and current use of contraceptives, as well as covariate variables are descriptively presented. Bivariable analysis with simple logistic regressions was carried out to perform the Crude Odds Ratio

which presented a 95% Confidence Interval (CI) to measure the variables' associations. Multiple logistic regression was used to examine relationships between women's empowerment and outcome variables adjusting for covariates variables regression with svy command to adjust for the DHS sampling design. A p-value of < 0.05 was interpreted as indicating statistical significance. Data were analyzed using STATA 14.

## Results

The total sample consisted of 5,011 women in Timor-Leste, 6,047 in Myanmar, 10,403 in Cambodia, 12,815 in Philippines and 31,495 in Indonesia by weighted sample size. The percentage of women currently using contraceptives was highest in Indonesia at 63.3%, followed by Cambodia 56.2%, Myanmar 52.4%, The Philippines 52.3% and the lowest in Timor-Leste at 27.5% (Table 1). The selected socio-demographic characteristics of the total sample as well as the covariates are displayed in Table 1.

Table 2 shows the proportion of women who were using contraceptive was higher in high labor force participation than low and middle in all countries, except for Indonesia. Women who disagreed with wife beating were more likely to use contraception compared to those who were more accepting in Myanmar and Timor-Leste, but the reverse was true in Cambodia, Indonesia and The Philippines. Women with medium decision-making power was more likely to utilize contraception In the Philippines, Myanmar, and Indonesia. In Timor-Leste, the proportion of women who now utilize contraception was higher among women with strong decision-making power, while in Cambodia, the proportion was higher among women with low decision-making power. Similarly, in Cambodia, Myanmar, and Timor-Leste, women with higher levels of knowledge were more likely to use contraception. However, in the Philippines and Indonesia, the proportion of women using contraception was higher among women with a medium knowledge level.

Table 3 shows the relationship between women's empowerment and current contraceptive use after controlling for the identified covariates. In this adjusted analysis, women's labor-force participation was significantly correlated with current contraceptive use in Cambodia, Indonesia, Myanmar, The Philippines, and Timor-Leste. Cambodian women with high and medium participation in the labor force had 2.01 and 1.97 greater odds of currently using contraception, compared with women with low participation. Indonesian women with high participation in the labor force had 1.09 greater odds of current contraceptive use compared with women with low participation. In Myanmar, women with high and medium participation in the labor force had 1.47 and 1.33 greater odds of currently using contraception compared with women with low participation. In The Philippines, women with high and medium participation in the labor force had 1.54 and 1.25 greater odds respectively of currently using contraception compared with women with low participation. In Timor-Leste, women with high and medium participation in the labor force had 1.62 and 1.24 greater odds of currently using contraception compared with women with low participation.

It was notable that women's disagreement with wife beating was not significantly correlated with contraceptive use after adjustment for covariates in any of the five countries. In addition, decision-making power only remained significantly correlated with current contraceptive use in Cambodia, where the odds of current contraceptive use was 0.82 in women with higher levels than in women with low levels of power. We also found that women's knowledge was only correlated with current contraceptive use in Cambodia and Myanmar. In Cambodia, women with high knowledge levels had 1.23 greater odds of current contraceptive use compared with women who had low knowledge. Similarly, women in Myanmar with high and medium knowledge levels were 1.48 and 1.45 greater odds to be using contraceptives, respectively.

**Table 1. Weighted characteristics of currently married women in the last five years preceding the survey in Cambodia, Indonesia, Myanmar, Philippines, and Timor-Leste.**

| Variables | Cambodia 2014 (N = 10,403) | | Indonesia 2017 (N = 31,495) | | Myanmar 2015–2016 (N = 6,047) | | Philippines 2017 (N = 12,815) | | Timor-Leste 2016 (N = 5,011) | |
|---|---|---|---|---|---|---|---|---|---|---|
| | N | % | N | % | N | % | N | % | N | % |
| **Current Contraceptive Use** | | | | | | | | | | |
| Currently not using | 4,555 | 43.8 | 11,552 | 36.7 | 2,877 | 47.6 | 6,118 | 47.7 | 3,632 | 72.5 |
| Currently using | 5,848 | 56.2 | 19,942 | 63.3 | 3,170 | 52.4 | 6,697 | 52.3 | 1,380 | 27.5 |
| **Key National Indicators** | | | | | | | | | | |
| CPR (Contraceptive Prevalence Rate) | - | 56 | - | 64 | - | 52 | - | 54 | - | 26 |
| TFR (Total Fertility Rate) | - | 2.6 | - | 2.3 | - | 2.2 | - | 2.9 | - | 3.7 |
| Unmet Need for Family Planning | - | 12.5 | - | 11 | - | 16 | - | 17 | - | 25 |
| **Women's Age** | | | | | | | | | | |
| 15–19 years | 409 | 3.9 | 627 | 2 | 172 | 2.8 | 391 | 3.1 | 136 | 2.7 |
| 20–24 years | 1,713 | 16.5 | 3,117 | 9.9 | 689 | 11.4 | 1,623 | 12.7 | 662 | 13.2 |
| 25–29 years | 2,062 | 19.8 | 5,183 | 16.5 | 1,045 | 17.3 | 2,438 | 19 | 1,091 | 21.8 |
| 30–34 years | 2,345 | 22.5 | 6,072 | 19.3 | 1,239 | 20.5 | 2,394 | 18.7 | 1,117 | 22.3 |
| 35–39 years | 1,339 | 12.9 | 6,380 | 20.3 | 1,176 | 19.4 | 2,319 | 18.1 | 655 | 13.1 |
| 40–44 years | 1,385 | 13.3 | 5,442 | 17.3 | 941 | 15.6 | 1,911 | 14.9 | 783 | 15.6 |
| 45–49 years | 1,151 | 11.1 | 4,674 | 14.8 | 785 | 13 | 1,739 | 13.6 | 566 | 11.3 |
| **Husband's education** | | | | | | | | | | |
| Incomplete primary education/none | 4,892 | 47 | 3,320 | 10.5 | 2,075 | 34.3 | 3,257 | 25.4 | 1,863 | 37.2 |
| Complete primary or some secondary | 4,229 | 40.7 | 14,718 | 46.7 | 3,431 | 56.7 | 5,548 | 43.3 | 1,036 | 20.7 |
| Completed secondary or higher | 1,281 | 12.3 | 13,456 | 42.7 | 541 | 8.9 | 4,010 | 31.3 | 2,113 | 42.2 |
| **Residence** | | | | | | | | | | |
| Urban | 1,563 | 15 | 15,126 | 48 | 1,524 | 25.2 | 5,658 | 44.1 | 1,545 | 30.8 |
| Rural | 8,840 | 85 | 16,368 | 52 | 4,523 | 74.8 | 7,157 | 55.9 | 3,466 | 69.2 |
| **Household Members** | | | | | | | | | | |
| <4 | 1,560 | 15 | 7,816 | 24.8 | 1,257 | 20.8 | 2,445 | 19.1 | 435 | 8.7 |
| ≥4 | 8,843 | 85 | 23,678 | 75.2 | 4,790 | 79.2 | 10,370 | 80.9 | 4,576 | 91.3 |
| **Wealth quintile** | | | | | | | | | | |
| Poorest | 2,021 | 19.4 | 5,543 | 17.6 | 1,305 | 21.6 | 2,708 | 21.1 | 885 | 17.7 |
| Poorer | 2,143 | 20.6 | 6,294 | 20 | 1,254 | 20.7 | 2,631 | 20.5 | 918 | 18.3 |
| Middle | 2,069 | 19.9 | 6,603 | 21 | 1,235 | 20.4 | 2,600 | 20.3 | 978 | 19.5 |
| Richer | 2,062 | 19.8 | 6,750 | 21.4 | 1,172 | 19.4 | 2,588 | 20.2 | 1,062 | 21.2 |
| Richest | 2,108 | 20.3 | 6,304 | 20 | 1,080 | 17.9 | 2,288 | 17.9 | 1,168 | 23.3 |
| **Husband's occupation** | | | | | | | | | | |
| Agricultural | 5,210 | 50.1 | 16,956 | 53.8 | 1,567 | 25.9 | 3,028 | 23.6 | 1,681 | 33.5 |
| Non-agricultural | 5,193 | 49.9 | 14,539 | 46.2 | 4,480 | 74.1 | 9,787 | 76.4 | 3,331 | 66.5 |
| **Difference in age between man and woman** | | | | | | | | | | |
| Woman older than man | 2,786 | 26.8 | 5,370 | 17 | 1,917 | 31.7 | 3,856 | 30.1 | 856 | 17.1 |
| Husband 0–4 years older | 4,375 | 42.1 | 12,383 | 39.3 | 2,373 | 39.2 | 5,009 | 39.1 | 1,782 | 35.6 |
| Husband 5–7 years older | 1,798 | 17.3 | 7,148 | 22.7 | 909 | 15 | 1,964 | 15.3 | 1,054 | 21 |
| Husband >7 years older | 1,444 | 13.9 | 6,593 | 20.9 | 848 | 14 | 1,987 | 15.5 | 1,320 | 26.3 |
| **Fertility intention** | | | | | | | | | | |
| Wants more children | 4,648 | 44.7 | 14,071 | 44.7 | 2,182 | 36.1 | 4,479 | 34.9 | 2,071 | 41.3 |
| Wants no more children | 5,421 | 52.1 | 16,319 | 51.8 | 3,669 | 60.7 | 7,494 | 58.5 | 1,408 | 28.1 |
| Undecided | 333 | 3.2 | 1,105 | 3.5 | 196 | 3.2 | 843 | 6.6 | 1,532 | 30.6 |
| **Number of Living Children** | | | | | | | | | | |

*(Continued)*

**Table 1.** (Continued)

| Variables | Cambodia 2014 (N = 10,403) | | Indonesia 2017 (N = 31,495) | | Myanmar 2015–2016 (N = 6,047) | | Philippines 2017 (N = 12,815) | | Timor-Leste 2016 (N = 5,011) | |
|---|---|---|---|---|---|---|---|---|---|---|
| | N | % | N | % | N | % | N | % | N | % |
| No Children | 1,014 | 9.7 | 2,420 | 7.7 | 689 | 11.4 | 1,042 | 8.1 | 313 | 6.3 |
| 1–2 | 5,356 | 51.5 | 20,385 | 64.7 | 3,325 | 55 | 6,490 | 50.6 | 1,685 | 33.6 |
| 3–4 | 2,999 | 28.8 | 7,525 | 23.9 | 1,533 | 25.3 | 3,641 | 28.4 | 1,588 | 31.7 |
| 5+ | 1,034 | 9.9 | 1,164 | 3.7 | 500 | 8.3 | 1,642 | 12.8 | 1,424 | 28.4 |
| **Knowledge of fertile period** | | | | | | | | | | |
| With correct knowledge | 2,410 | 23.2 | 7,286 | 23.1 | 392 | 6.5 | 3,391 | 26.5 | 544 | 10.8 |
| With incorrect knowledge | 7,992 | 76.8 | 24,209 | 76.9 | 5,655 | 93.5 | 9,424 | 73.5 | 4,468 | 89.2 |
| **Heard about FP on the radio in the past few months** | | | | | | | | | | |
| No | 6,488 | 62.4 | 28,661 | 91 | 5,073 | 83.9 | 8,019 | 62.6 | 4,128 | 82.4 |
| Yes | 3,915 | 37.6 | 2,833 | 9 | 974 | 16.1 | 4,796 | 37.4 | 883 | 17.6 |
| **Heard about FP on the TV in the past few months** | | | | | | | | | | |
| No | 5,122 | 49.2 | 13,319 | 42.3 | 4,500 | 74.4 | 4,640 | 36.2 | 3,987 | 79.6 |
| Yes | 5,280 | 50.8 | 18,175 | 57.7 | 1,547 | 25.6 | 8,175 | 63.8 | 1,024 | 20.4 |
| **Heard about FP on the newspaper in the past few months** | | | | | | | | | | |
| No | 8,766 | 84.3 | 27,844 | 88.4 | 5,000 | 82.7 | 10,298 | 80.4 | 4,539 | 90.6 |
| Yes | 1,637 | 15.7 | 3,651 | 11.6 | 1,047 | 17.3 | 2,518 | 19.6 | 473 | 9.4 |

**Table 2. Distribution of current contraceptive use by women's empowerment indicators in Cambodia, Indonesia, Myanmar, Philippines, and Timor-Leste.**

| Variables | Cambodia (N = 10,403) | | Indonesia (N = 31,495) | | Myanmar (N = 6,047) | | Philippines (N = 12,815) | | Timor-Leste (N = 5,011) | |
|---|---|---|---|---|---|---|---|---|---|---|
| | N | Currently using (%) | N | Currently using (%) | N | Currently using (%) | N | Currently using (%) | N | Currently using (%) |
| **Labor force participation** | | | | | | | | | | |
| Low | 3,630 | 46.3 | 11,119 | 65.3 | 1,915 | 50.1 | 4,288 | 52.1 | 1,681 | 23.2 |
| Medium | 3,588 | 60.7 | 10,166 | 62.3 | 1,973 | 51.8 | 4,046 | 52.2 | 1,762 | 27 |
| High | 3,184 | 62.5 | 10,210 | 62.2 | 2,159 | 55.1 | 4,481 | 52.4 | 1,568 | 32.8 |
| **Disagreement to justification toward wife beating** | | | | | | | | | | |
| Low | 3,497 | 56.9 | 9,544 | 63.2 | 2,078 | 49.8 | 3,754 | 53.2 | 1,654 | 27.4 |
| Medium | 3,656 | 57.5 | 11,947 | 64.3 | 2,077 | 53.3 | 4,348 | 53 | 1,718 | 27.4 |
| High | 3,250 | 54 | 10,004 | 62.3 | 1,892 | 54.3 | 4,713 | 50.8 | 1,639 | 27.9 |
| **Decision-making power** | | | | | | | | | | |
| Low | 3,062 | 58 | 11,119 | 63.2 | 2,096 | 53.1 | 4,258 | 50.1 | 1,628 | 25 |
| Medium | 3,647 | 55.7 | 10,014 | 64.1 | 2,069 | 53.2 | 4,383 | 54 | 1,618 | 28.4 |
| High | 3,693 | 55.2 | 10,361 | 62.7 | 1,882 | 50.8 | 4,174 | 52.6 | 1,765 | 29.1 |
| **Women's knowledge level** | | | | | | | | | | |
| Low | 3,664 | 55.1 | 11,435 | 64.2 | 2,038 | 44.6 | 3,646 | 52.3 | 1,674 | 25.2 |
| Medium | 3,628 | 56.3 | 11,090 | 65.2 | 2,035 | 55.7 | 4,399 | 55.5 | 1,662 | 28.4 |
| High | 3,111 | 57.4 | 8,970 | 59.8 | 1,975 | 57.1 | 4,770 | 49.3 | 1,675 | 29 |

**Table 3. Adjusted odds ratios of the association between women's empowerment indicators and currently contraceptive use adjusted for covariates[a].**

| | Cambodia | | Indonesia | | Myanmar | | Philippines | | Timor-Leste | |
|---|---|---|---|---|---|---|---|---|---|---|
| | AOR | 95% CI | AOR | 95% CI | AOR | 95% CI | AOR | 95% CI | AOR | 95% CI |
| **Labor force participation** | | | | | | | | | | |
| Low | Ref | | Ref | | Ref | | Ref | | Ref | |
| Medium | 1.97*** | 1.73–2.24 | 1.06 | 0.97–1.16 | 1.33*** | 1.13–1.56 | 1.25** | 1.07–1.45 | 1.24* | 1.00–1.55 |
| High | 2.01*** | 1.73–2.34 | 1.09* | 1.00–1.20 | 1.47*** | 1.25–1.74 | 1.54*** | 1.29–1.82 | 1.62*** | 1.32–1.99 |
| **Disagreement to justification toward wife beating** | | | | | | | | | | |
| Low | Ref | | Ref | | Ref | | Ref | | Ref | |
| Medium | 1.00 | 0.87–1.15 | 1.01 | 0.92–1.10 | 1.10 | 0.94–1.28 | 1.07 | 0.90–1.28 | 0.93 | 0.75–1.14 |
| High | 0.89 | 0.77–1.04 | 1.00 | 0.92–1.09 | 1.08 | 0.92–1.28 | 0.94 | 0.76–1.17 | 0.90 | 0.72–1.12 |
| **Decision making power** | | | | | | | | | | |
| Low | Ref | | Ref | | Ref | | Ref | | Ref | |
| Medium | 0.82** | 0.72–0.93 | 1.04 | 0.96–1.13 | 1.03 | 0.88–1.21 | 1.15 | 0.99–1.34 | 1.11 | 0.90–1.37 |
| High | 0.82** | 0.72–0.94 | 0.93 | 0.85–1.02 | 0.93 | 0.78–1.10 | 0.99 | 0.84–1.16 | 1.12 | 0.92–1.36 |
| **Women's knowledge level** | | | | | | | | | | |
| Low | Ref | | Ref | | Ref | | Ref | | Ref | |
| Medium | 1.09 | 0.93–1.28 | 1.04 | 0.95–1.13 | 1.45*** | 1.22–1.72 | 1.12 | 0.94–1.34 | 0.99 | 0.79–1.24 |
| High | 1.23* | 1.03–1.46 | 0.96 | 0.86–1.06 | 1.48*** | 1.21–1.80 | 1.08 | 0.89–1.32 | 0.96 | 0.73–1.25 |
| **Observations** | 10,064 | | 30,105 | | 6,100 | | 13,219 | | 5,004 | |
| **Population Size** | 10,403 | | 31,495 | | 6047 | | 12815 | | 5011 | |

[a]Adjusted for women's age, residence, socioeconomic status, number of household member, husband's education and employment, age difference between husband and wife, fertility intention, knowledge of fertile period, and media sources of family planning (radio, TV, and newspaper).

AOR = adjusted odds ratio

*** p<0.001

** p<0.01

* p<0.05.

## Discussion

We analyzed the association between women's empowerment and their contraceptive use in Indonesia, Cambodia, the Philippines, Myanmar, and Timor-Leste. In all five nations, labor force participation was positively associated with contraceptive use, whereas disagreement with wife beating had no association after adjusting for covariates. Interestingly, women's decision-making power was negatively correlated with contraceptive use in Cambodia, while knowledge level was positively correlated with current use in both Cambodia and Myanmar. The findings suggest that women's empowerment and contraceptive use is a complex relationship that may be further influenced by factors other than those measured in the survey. In addition, it is unclear what levels of contraceptive use are appropriate for these countries to aim to achieve.

Strong evidence of women's empowerment and contraceptive use among ASEAN member states needs to be examined in a regional integration context. Regional commitment to bolster women's empowerment remains a pressing issue in societies where women are under-represented in many sectors [37]. Recent data from 67 developing countries showed that women's empowerment was associated with positive outcomes for health care services [15]. This finding is similar with our study in that certain indicators of women's empowerment remain critical in attaining women's reproductive health choices, population control and health goals.

The present study found that women with high labor force participation had 1.09–2.01 higher odds of using contraceptives in Indonesia, Cambodia, The Philippines, Myanmar, and

Timor-Leste. In the same vein, Kabeer (2012) noted that women's empowerment can be advanced through labor force participation [27]. Giving women opportunities to to make their own decisions may improve the chances of seeking employment and further strengthen their capacity to negotiate within the household and learn new skills [38].

Interestingly, there was no correlation between high levels of disagreement with wife beating and contraceptive use after adjusting for covariates which conflicts with previous research. In a study of African women by Do and Kurimoto (2012), a positive association was reported between empowered attitudes toward violence among women and contraceptive use [13]. In addition, a national survey among Bangladeshi women indicated that high rejection of justification with wife beating was associated with 1.19 times greater odds of contraceptive use [39].

Our findings only found significant results in Cambodia in terms of decision making. Similar to previous Cambodian studies, significant increases in contraceptive use occurred in women's involvement in household decision making [40]. Our results do not support findings from previous studies on women's empowerment and decision making. Women who had high and medium decision making were less likely to use contraception, compared to women who had low decision-making power. In contrast, Patrikar, Basannar, and Seema Sharma's (2014) study confirmed that women with high levels of decision-making power played a significant role in utilizing one family planning method [41]. Another study also reported that women's involvement in household decision making was positively correlated with a 28% increase in the likelihood of contraceptive use [9]. As these studies were not from the ASEAN region, it is possible that this situation is highly contextual and complex in nature.

The present findings seem to be consistent with previous research that found significant relationships between women's knowledge levels and use of contraception in Cambodia and Myanmar. A study conducted among reproductive-aged women in Ghana showed that high levels of knowledge and awareness of contraception were predictors of using contraception [42]. Another similar study in Rwanda found that women's knowledge levels were significant determinants of contraceptive use among women respondents of reproductive age [43].

This study showed that labor force participation was positively correlated with current use of contraceptives for women in the five ASEAN countries. The female labor force participation (FLFP) rate varied considerably across the countries from 81% in Cambodia, 51% in Indonesia and Myanmar, 50% in Philippines and 25% in Timor Leste among the female population aged more than 15 years old [44]. The relationship between women in the labor force and contraceptive use is open to investigation. Firstly, this association may be related to affordability as working women may have resources to access contraception. Furthermore, accessible contraceptive services have been improved and expanded. Policy documents from Indonesia [45], The Philippines [46] and Timor Leste [47] indicate a broader expansion and more affordability of family planning services. In addition, participation in the labor market may require women to delay pregnancies and temporarily use contraceptives.

As mentioned above, disagreement with wife beating was not correlated with contraceptive use in this study after adjusting for covariates. Although ASEAN member states have made numerous commitments to minimize violence against women, rates of violence remain high [48]. This may be due to patriarchal gender norms and entrenched views surrounding the tolerance of violence against women within these countries. Studies in some developing countries (17 sub-Saharan countries) suggest that women more often justify violence against other women than do men [49].

The decision-making power in a household is relative and varies across countries, cultures and social norms [50]. Cambodia, like most Southeast Asian countries, remains a hierarchical society with strong ideas of power and status. In general, attitudes toward gender roles continue to favor women as housewives and men as breadwinners. Women play a role in

decision-making in the home, but they are significantly underrepresented in decision-making processes outside the home [51]. Indonesian society has a strong patriarchal culture [52] similar to Myanmar that struggled to embed the principles of women's rights and gender equality and shared decision-making power within the family [53]. Conversely, the Philippines is reported to be one of the gender equal countries in the Asian region [54]. Despite this, in relation to decision-making power, results are varied between Filipino husbands and wives about whether and how to practice contraception [55]. As the youngest ASEAN member state, a study regarding women of Timor Leste highlighted that husband, intra-spousal relationships between couples and norms of society had impacted their decision making power [56]. Another study conducted outside of the ASEAN region revealed urban Ethiopian women showed high propensity to have better power to make decisions on using modern types of contraceptive than rural Ethiopian women [57]. In marital relationships, women may be more economically dependent on their partners [58] and require permission to obtain health care or purchase contraceptives. In this context, it is notable that the decision-making power of women in Indonesia, Myanmar, the Philippines, and Timor-Leste was not correlated with contraceptive use after adjusting for covariates.

In Cambodia, higher educated women have a higher CPR than less educated women. Higher educated women are more informed, prefer fewer children, and face a higher opportunity cost when it comes to having children [40]. According to the Myanmar DHS 2015–2016, almost all married women (98.5%) were familiar with some form of modern contraception. The CPR of married women rises as their educational level rises. The high level of reproductive health knowledge among Myanmar women may be due to a government program called Youth Information Corners (YICs) run by the Ministry of Health's Division of Health Education, which works with UNFPA to promote reproductive health knowledge among adolescents [59, 60]. Different from the results in Cambodia and Myanmar, women's knowledge levels were not significantly correlated with contraceptive use in Indonesia, the Philippines and Timor-Leste, despite their relatively high TFRs (2.36, 2.92 and 5.50 respectively). This is surprising as in the Philippines and Timor-Leste, our view is that religion, culture and family values might exercise strong direct influence on fertility intention. Both countries' populations profess allegiance to the Roman Catholic faith, 72% in the Philippines and 98% in Timor-Leste [31, 34]. This phenomenon may warrant further scrutiny, particularly to the extent that religion and associated values affect contraceptive use and other family planning. Furthermore, one study noted that cultural factors in the Philippines worked against a decline in wanted fertility as most parents believed that having multiple children was better than only having one child [61]. In the case of Indonesia, the country has had a long-standing and solid family planning program in place since 1970. In addition, Indonesia's establishment of the Village Midwife Program in 1989, integrating provision of information and counseling, may positively influence women's knowledge regarding the use of contraception [62]. Although women are knowledgeable about reproductive health, they lack cultural power. Several groups in Indonesia still believe that contraception should be prohibited, and women who do not follow these rules will be considered disobedient [63, 64].

The limitations of this study include that only five countries in the ASEAN region were included. Across the five countries, and certainly more broadly across the region, there is great cultural and social variation so findings cannot be generalized. We did exclude never-married women from our analysis, which may have underestimated the association between women's empowerment and contraceptive use. However, the number of unmarried women reporting contraceptive use was low (0.21% in Indonesia, 0.39% in Cambodia, 0.07% in Myanmar, 1.58% in Philippines, and 0.15% in Timor-Leste), so the underestimation may also have been low. Empowerment is only one component of contraceptive use. Hence, there may be other

factors that could impact on empowerment and contraceptive use and a need for more studies. Another issue is that this study used a correlational design and cannot assume causality in any of the observations. We did employ a large number of covariates and that does raise the potential of multicollinearity. Furthermore we changed husband's age factor into age difference with wife to avoid multicollinearity.

## Conclusion

This study has provided insight into women's empowerment and contraceptive use in five ASEAN countries. Women's empowerment, especially in terms of labor-force participation as a measure of resources, was found to be positively associated with contraceptive use, while agency component of women's empowerment showed a complex association with contraceptive use. Structured regional policies are required to ensure that gender equality, especially in work opportunities, is deployed as a development priority in ASEAN countries. Ensuring that no woman is left behind will be a major challenge moving forward with policy implementation at both national and family levels. Irrespective of the approach taken, it will be essential to understand that a woman's economic opportunities are vital to facilitating choice. The creation of job opportunities for ASEAN women may therefore, speed the necessary progress toward effective contraceptive use. Innovations to accelerate the engagement of women and improve empowerment should be tested in the future.

## Author Contributions

**Conceptualization:** Ferry Efendi, Susy Katikana Sebayang, Erni Astutik, Sonia Reisenhofer, Lisa McKenna.

**Formal analysis:** Ferry Efendi, Susy Katikana Sebayang, Erni Astutik.

**Methodology:** Ferry Efendi, Susy Katikana Sebayang, Erni Astutik, Sonia Reisenhofer, Lisa McKenna.

**Writing – original draft:** Ferry Efendi, Susy Katikana Sebayang, Erni Astutik, Sonia Reisenhofer, Lisa McKenna.

**Writing – review & editing:** Ferry Efendi, Susy Katikana Sebayang, Erni Astutik, Sonia Reisenhofer, Lisa McKenna.

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
