## [Decision Letter · Decision Letter 0]

13 Sep 2022

PONE-D-22-02451Women’s Empowerment and Contraceptive Use: Recent Evidence from ASEAN CountriesPLOS ONE

Dear Dr. Efendi,

Thank you for submitting your manuscript to PLOS ONE. After careful consideration, we feel that it has merit but does not fully meet PLOS ONE’s publication criteria as it currently stands. Therefore, we invite you to submit a revised version of the manuscript that addresses the points raised during the review process.

 After my review of the manuscript, it is still need to improve and worths giving a chance for revision. So, I made the decision "Major Revision".Please review the specific comments of each reviewer and respond in detail.In addition, the discussion session should be strengthened. 

We look forward to receiving your revised manuscript.

Kind regards,

Thae Maung Maung, MBBS, MSc (International Health), PhD

Academic Editor

PLOS ONE

Journal Requirements:

” F.E. received funding support from the Ministry of Research, Technology and Higher Education of Indonesia during his post-doctoral degree. The funding source was not involved in study design, data collection, analysis or interpretation; in the writing of this report; or in the decision to submit the article for publication.”

Reviewers' comments:

Reviewer's Responses to Questions

**Comments to the Author**

1. Is the manuscript technically sound, and do the data support the conclusions?

Reviewer #1: Partly

Reviewer #2: Yes

2. Has the statistical analysis been performed appropriately and rigorously? 

Reviewer #1: No

Reviewer #2: Yes

3. Have the authors made all data underlying the findings in their manuscript fully available?

Reviewer #1: Yes

Reviewer #2: Yes

4. Is the manuscript presented in an intelligible fashion and written in standard English?

Reviewer #1: Yes

Reviewer #2: Yes

5. Review Comments to the Author

Reviewer #1: Comments on PONE-D-22-02451

Studies of the relationship of women’s empowerment and contraceptive use are important, and needed generally and in countries of the ASEAN region. The authors of this manuscript leverage publicly available Demographic and Health Survey data for several ASEAN countries to examine the relationship of four variables they use to operationalize women’s empowerment (labor force participation; disagreement with reasons for wife beating; decision-making power over household issues; and ‘knowledge level’) with women’s current contraceptive use. I hope my comments below contribute to strengthening the contribution of this manuscript to the field.

Background

1. The background section would benefit from a clear definition of women’s empowerment, with reference to the appropriate theoretical literature. I recommend Kabeer (1999) as a seminal reference to cite.

2. This reference will help the authors distinguish conceptually between the ‘resources’ measures of empowerment they include (e.g., labor force participation) and the ‘agency’ measures they include (e.g., attitudes about wife beating; influence in decisions, and knowledge). Thus, the variables they use to operationalize ‘women’s empowerment’ should be conceptualized according to the ‘resources’ and ‘agency’ components of women’s empowerment.

3. Relevant references are missing from the background and should be added. A key reference is a systematic review of the literature on women’s agency and contraceptive use, by Laurie James-Hawkins and colleagues: James-Hawkins, L., Peters, C., VanderEnde, K., Bardin, L., & Yount, K. M. (2018). Women’s agency and its relationship to current contraceptive use in lower-and middle-income countries: A systematic review of the literature. Global Public Health, 13(7), 843-858. More nuanced discussion of the empirical literature on women’s empowerment and contraceptive use also may be useful.

Methods

1. More information about each sample is needed, for example response rates and any missing data on variables of interest and how those missing data are handled in the analysis.

2. The variable ‘knowledge level’ needs to be defined more clearly in the methods and throughout the manuscript (knowledge level of what?).

3. The variable for ‘contraceptive use’ also needs to be defined more clearly in the methods and throughout the manuscript (e.g., what methods were asked about in each survey; were the methods asked about the same across surveys? What does ‘current’ mean, e.g., at the time of the survey?)

4. Why is age included separately in the ‘conceptual framework’ (which anyway should go into the background, not the methods section)? Is age rather a covariate?

5. Why is women’s schooling not included as a resource for their empowerment?

6. The variables for women’s ‘agency’ (attitudes about wife beating, decision-making influence, knowledge) need to be described in much more detail. How many items are available each construct? Are the number of items the same across countries (they should be). What are the response options for each construct? How response options were coded/recoded for the analysis? How specifically were these variables constructed for the analysis? What does ‘low, medium, high’ mean for each variable, and is this an appropriate way to model these measures (what is the theoretical rationale for doing so?)

7. The Analysis section should be separate from the Variables section in the Methods, and much more detail should be provided on the a) descriptive analyses performed in the pooled and country-specific samples, b) process for constructing each of the agency scales/variables and assessing their item and scale distributions and alpha reliability within and across countries, and c) sequential regression analysis (unadjusted and adjusted models in the pooled and/or stratified samples). Also if women’s LFP is a resource for empowerment, is a hierarchical model (unadjusted models, resources for empowerment, resources and agency for empowerment to assess mediation) more appropriate on theoretical grounds? Also, how was the multi-stage cluster sample design handled for each country?

Results and Discussion

1. Without the above detail, it is difficult to interpret the accuracy or reasonableness of the findings.

2. After the above methods issues are sorted, I recommend that the Discussion section also include a thoughtful discussion of the measures of empowerment (resources and/or agency) that were not included but that may be relevant for all countries or for specific countries. For example, is women’s freedom of movement or lack thereof relevant for some ASEAN countries if not for others? What are the implications for the results of leaving out important cross-country or country-specific measures of women’s resources and/or agency? If you could, what alternative measures might you use for constructs of agency that were not associated with contraceptive use in the final models? Finally, what are the potential pitfalls of recommending that women’s labor force participation is important for their empowerment without certain qualifications (e.g., implications for their time-use agency; triple burden of paid labor, unpaid domestic labor, and unpaid care work)

Thank you for the opportunity to review this manuscript. I hope the above comments are helpful.

Reviewer #2: In figure(1), why Age in a separate box as age is a covariate and could not see the analysis with age. Outcome is Current use of contraception and no need to present "not use"

The working definition for tertiles should be mentioned.

One sentence for Analysis is not enough.

In Table (1), no need to present "currently not using", the row for No in "Knowledge of fertile period", "heard about FP on radio", "TV", "newspaper" and label should be "women" in stead of "mother".

6. PLOS authors have the option to publish the peer review history of their article (what does this mean?). If published, this will include your full peer review and any attached files.

Reviewer #1: No

Reviewer #2: No

---

## [Author Response · Author response to Decision Letter 0]

15 Nov 2022

AUTHORS’ RESPONSES TO REVIEWERS

Academic Editor

No Reviewer’s Comments Authors’ Responses

1 Please include the following items when submitting your revised manuscript:

 Many thanks, we have prepared the three files as requested

2 1. Please ensure that your manuscript meets PLOS ONE's style requirements, including those for file naming. The PLOS ONE style templates can be found at

 Many thanks, we have ensured that the manuscript meets PLOS ONE's style requirements

3 2. Thank you for stating in your Funding Statement:

Please include your amended Funding Statement within your cover letter. We will change the online submission form on your behalf. ” F.E. received funding support from the Ministry of Research, Technology and Higher Education of Indonesia during his post-doctoral degree. The funding source was not involved in study design, data collection, analysis or interpretation; in the writing of this report; or in the decision to submit the article for publication.”

 Reviewer 1 

5 1. Is the manuscript technically sound, and do the data support the conclusions?

Reviewer #1: Partly

Reviewer #2: Yes

 Thank you for the comment

6 2. Has the statistical analysis been performed appropriately and rigorously?

Reviewer #1: No

Reviewer #2: Yes

 Thank you for the comment

7 3. Have the authors made all data underlying the findings in their manuscript fully available?

Reviewer #1: Yes

Reviewer #2: Yes

 Thank you for the comment

8 4. Is the manuscript presented in an intelligible fashion and written in standard English?

Reviewer #1: Yes

Reviewer #2: Yes

 Thank you for the comment

9 5. Review Comments to the Author

Reviewer #1: Comments on PONE-D-22-02451

Studies of the relationship of women’s empowerment and contraceptive use are important, and needed generally and in countries of the ASEAN region. The authors of this manuscript leverage publicly available Demographic and Health Survey data for several ASEAN countries to examine the relationship of four variables they use to operationalize women’s empowerment (labor force participation; disagreement with reasons for wife beating; decision-making power over household issues; and ‘knowledge level’) with women’s current contraceptive use. I hope my comments below contribute to strengthening the contribution of this manuscript to the field. Thank you for the comment

10 Background

1. The background section would benefit from a clear definition of women’s empowerment, with reference to the appropriate theoretical literature. I recommend Kabeer (1999) as a seminal reference to cite. Thank you very much. We add the reference in the background.

“Women's empowerment according to Kabir (1999) is the ability of women to make life choices which combines 3 interrelated dimensions including: resources (access, material and human and social resources); agency (decision-making processes, less measurable manifestations of agency such as negotiation, deception, and manipulation); and achievement (welfare outcomes) (Kabeer, 1999a; Kabeer, 1999b).”

1. Kabeer, N. (1999). Resources, agency, achievements: Reflections on the measurement of women's empowerment. Development and change, 30(3), 435-464.

2. Kabeer, N. (1999). The conditions and consequences of choice: reflections on the measurement of women's empowerment (Vol. 108, pp. 1-58). Geneva: UNRISD.

11 2. This reference will help the authors distinguish conceptually between the ‘resources’ measures of empowerment they include (e.g., labor force participation) and the ‘agency’ measures they include (e.g., attitudes about wife beating; influence in decisions, and knowledge). Thus, the variables they use to operationalize ‘women’s empowerment’ should be conceptualized according to the ‘resources’ and ‘agency’ components of women’s empowerment. Thank you very much. We add the reference in the background.

“The role of women's empowerment is measured by variables, namely labor force participation, disagreement to justification toward wife beating, decision-making power over household issues, and knowledge level (Phan, 2015;Sebayang, Efendi, and Astutik, 2017). Labor force participation is part of component of the resources. Disagreement to justification toward wife beating, decision-making power over household issues, and knowledge level variables are part of the agency component of women's empowerment roles (Kabeer, 1999a; Kabeer, 1999b).”

1. Phan L. Measuring women’s empowerment at household level using DHS data of four Southeast Asian countries. Soc Indic Res. 2016;126(1):359–78.

2. Sebayang SK, Efendi F, Astutik E. Women’s empowerment and the use of antenatal care services: analysis of demographic health surveys in five Southeast Asian countries. Women Health [Internet]. 2019 Apr;1–17. Available from: https://www.tandfonline.com/doi/full/10.1080/03630242.2019.1593282

3. Kabeer, N. (1999). Resources, agency, achievements: Reflections on the measurement of women's empowerment. Development and change, 30(3), 435-464.

4. Kabeer, N. (1999). The conditions and consequences of choice: reflections on the measurement of women's empowerment (Vol. 108, pp. 1-58). Geneva: UNRISD.

12 3. Relevant references are missing from the background and should be added. A key reference is a systematic review of the literature on women’s agency and contraceptive use, by Laurie James-Hawkins and colleagues: James-Hawkins, L., Peters, C., VanderEnde, K., Bardin, L., & Yount, K. M. (2018). Women’s agency and its relationship to current contraceptive use in lower-and middle-income countries: A systematic review of the literature. Global Public Health, 13(7), 843-858. More nuanced discussion of the empirical literature on women’s empowerment and contraceptive use also may be useful. Thank you very much. We have added the reference in the background

“Research shows that there is a positive relationship between women's empowerment and reproductive health. Women's empowerment is associated with the use of female contraceptives in low- and middle-income countries. The results of the study also mention that understanding the causal impact of women's agency on contraceptive use, and will help inform policies and programs to increase contraceptive use (James-Hakins et al, 2018).”

James-Hawkins, L., Peters, C., VanderEnde, K., Bardin, L., & Yount, K. M. (2018). Women’s agency and its relationship to current contraceptive use in lower-and middle-income countries: A systematic review of the literature. Global Public Health, 13(7), 843-858.

 Methods

1. More information about each sample is needed, for example response rates and any missing data on variables of interest and how those missing data are handled in the analysis. Thank you very much, we add the information in the methods.

“We used complete cases for all analysis. We used complete cases for all analysis. The total number of respondents were 10,403 for Cambodia, 31,495 for Indonesia, 6,047 for Myanmar, 12,815 for The Philippines, and 5,011 for Timor-Leste. DHS data represent a response rate of 98% of women for Indonesia, Cambodia, and The Philippines, 97% of women for Timor Lester and 96% of women for Myanmar.”

1. National Institute of Statistics. Cambodia Demographic and Health Survey 2014 [Internet]. 2015 [cited 2022 Oct 8]. Available from: https://dhsprogram.com/pubs/pdf/fr312/fr312.pdf

2. National Population and Family Planning Board. Indonesia Demographic and Health Survey 2017 [Internet]. 2018 [cited 2022 Oct 8]. Available from: https://dhsprogram.com/pubs/pdf/FR342/FR342.pdf

3. Statistics P, City Q. Philippines National Demographic and Health Survey 2017 Key Indicators Report. 2018; 

4. General Directorate of Statistics Ministry of Planning and Finance and Ministry of Health. Timor-Leste Demographic and Health Survey 2016 [Internet]. 2017 [cited 2022 Oct 8]. Available from: https://www.dhsprogram.com/pubs/pdf/FR329/FR329.pdf

5. Ministry of Health and Sports, Nay Pyi Taw M. MYANMAR DEMOGRAPHIC AND HEALTH SURVEY 2015-2016 [Internet]. 2017 [cited 2022 Oct 8]. Available from: https://dhsprogram.com/pubs/pdf/FR324/FR324.pdf

 2. The variable ‘knowledge level’ needs to be defined more clearly in the methods and throughout the manuscript (knowledge level of what?).

 Thank you very much, we added the information in the methods. 

“The last component on women’s knowledge includes access to media (newspaper, radio, and television) and the highest level of formal educational. Access to media was divided to 3 categories, namely not at all, some of media, all of media. the highest level of formal educational was categorized into no education, primary, secondary and higher education.”

 3. The variable for ‘contraceptive use’ also needs to be defined more clearly in the methods and throughout the manuscript (e.g., what methods were asked about in each survey; were the methods asked about the same across surveys? What does ‘current’ mean, e.g., at the time of the survey?) Thank you very much. We added the information in the methods

“Currently use of contraception was measured of percentage of women who currently use specific methods. Currently use the specific method (v312 = x), after being asked whether they or their partner are currently doing something or using any method to delay or avoid getting pregnant. Women who say they are pregnant are coded as not currently using any method.”

1. DHS. The DHS Program - DHS Survey Indicators - Family Planning [Internet]. [cited 2022 Oct 8]. Available from: https://dhsprogram.com/data/dhs-survey-indicators-family-planning.cfm

2. DHS. Current Use of Contraceptive Methods [Internet]. Available from: https://dhsprogram.com/data/Guide-to-DHS-Statistics/Current_Use_of_Contraceptive_Methods.htm

 4. Why is age included separately in the ‘conceptual framework’ (which anyway should go into the background, not the methods section)? Is age rather a covariate? Thank you very much. Age is one of our covariate variables

 5. Why is women’s schooling not included as a resource for their empowerment? Thank you very much. Women’s schooling was included in the component of women knowledge level. We added the information in the methods.

The last component on women’s knowledge includes access to media (newspaper, radio, and television) and the highest level of formal educational. Access to media was divided to 3 categories, namely not at all, some of media, all of media. the highest level of formal educational was categorized into no education, primary, secondary and higher education.

 6. The variables for women’s ‘agency’ (attitudes about wife beating, decision-making influence, knowledge) need to be described in much more detail. How many items are available each construct? Are the number of items the same across countries (they should be). What are the response options for each construct? How response options were coded/recoded for the analysis? How specifically were these variables constructed for the analysis? What does ‘low, medium, high’ mean for each variable, and is this an appropriate way to model these measures (what is the theoretical rationale for doing so?) Thank you very much. We added the information in the methods. The number of items were the same across countries. 

“We used Principal Component Analysis (PCA) to create empowerment components. PCA is adopted for creating women empowerment proxies in order to investigate the independent effects of women empowerment on certain health outcome (Sebayang, et al, 2020). These four components of women’s empowerment were calculated from 17 indicators from the DHS surveys using a principal component analysis. Components of labor force participation include six aspects: work in the last 12 months, for whom women work, women’s occupation, types of payment, working all years and earning more than husband. The second component of attitudes toward wife beating includes five aspects: neglecting children, going places without husband’s permission, arguing, refusing sex and burning food. The third component on decision making includes own health care, household purchasing, visiting family and husband’s earnings. “

 7. The Analysis section should be separate from the Variables section in the Methods, and much more detail should be provided on the a) descriptive analyses performed in the pooled and country-specific samples, b) process for constructing each of the agency scales/variables and assessing their item and scale distributions and alpha reliability within and across countries, and c) sequential regression analysis (unadjusted and adjusted models in the pooled and/or stratified samples). Also if women’s LFP is a resource for empowerment, is a hierarchical model (unadjusted models, resources for empowerment, resources and agency for empowerment to assess mediation) more appropriate on theoretical grounds? Also, how was the multi-stage cluster sample design handled for each country? Thank you very much. We revised it and we added information in methods

“The data analysis took into account the complex survey sampling design, which produced the weighted results. Respondents’ characteristics, the component of women’s empowerment, and Current use of Contraceptives as well as covariate variables were descriptively presented. The bivariate analysis with simple logistic regressions was carried out to perform the Crude Odds Ratio. Multiple logistic regression was used to examine the relationship between women’s empowerment and outcome variables adjusting for covariates variables regression with svy command to adjust for the DHS sampling design. Data were analyzed by STATA 14. “

 2. After the above methods issues are sorted, I recommend that the Discussion section also include a thoughtful discussion of the measures of empowerment (resources and/or agency) that were not included but that may be relevant for all countries or for specific countries. For example, is women’s freedom of movement or lack thereof relevant for some ASEAN countries if not for others? What are the implications for the results of leaving out important cross-country or country-specific measures of women’s resources and/or agency? If you could, what alternative measures might you use for constructs of agency that were not associated with contraceptive use in the final models? Finally, what are the potential pitfalls of recommending that women’s labor force participation is important for their empowerment without certain qualifications (e.g., implications for their time-use agency; triple burden of paid labor, unpaid domestic labor, and unpaid care work)

Thank you for the opportunity to review this manuscript. I hope the above comments are helpful.

 Thank you very much for this positive feedback, we have add more elaboration on the discussion section. 

 Reviewer #2: In figure(1), why Age in a separate box as age is a covariate and could not see the analysis with age. Thank you very much. Age is covariate variables. We revise the figure

“These covariates included age (categorized into five-year groups, namely 15-19, 20-24, 25-29, 30-34, 35-39, 40-44, 45-49 years old),…”

 Outcome is Current use of contraception and no need to present "not use" Thank you very much, we added some information in the methods

“Currently use of contraception was measured of percentage of women who currently use specific methods. Currently use the specific method (v312 = x), after being asked whether they or their partner are currently doing something or using any method to delay or avoid getting pregnant. Women who say they are pregnant are coded as not currently using any method.”

 The working definition for tertiles should be mentioned. Thank you very much, we added some information in the methods

“We used Principal Component Analysis (PCA) to create empowerment components. PCA is adopted for creating women empowerment proxies in order to investigate the independent effects of women empowerment on certain health outcome (Astutik, Erni, et al, 2020). These four components of women’s empowerment were calculated from 17 indicators from the DHS surveys using a principal component analysis. Components of labor force participation include six aspects: work in the last 12 months, for whom women work, women’s occupation, types of payment, working all years and earning more than husband. The second component of attitudes toward wife beating includes five aspects: neglecting children, going places without husband’s permission, arguing, refusing sex and burning food. The third component on decision making includes own health care, household purchasing, visiting family and husband’s earnings. “

 One sentence for Analysis is not enough.

 Thank you very much, we added some information in the methods

“The data analysis took into account the complex survey sampling design, which produced the weighted results. Respondents’ characteristics, the component of women’s empowerment, and Current use of Contraceptives as well as covariate variables were descriptively presented. The bivariate analysis with simple logistic regressions was carried out to perform the Crude Odds Ratio. Multiple logistic regression was used to examine the relationship between women’s empowerment and outcome variables adjusting for covariates variables regression with svy command to adjust for the DHS sampling design. Data were analyzed by STATA 14.”

 In Table (1), no need to present "currently not using", the row for No in "Knowledge of fertile period", "heard about FP on radio", "TV", "newspaper" and label should be "women" in stead of "mother". Thank you very much, we have revised it. 

 6. PLOS authors have the option to publish the peer review history of their article (what does this mean?). If published, this will include your full peer review and any attached files.

Do you want your identity to be public for this peer review? For information about this choice, including consent withdrawal, please see our Privacy Policy.

Reviewer #1: No

Reviewer #2: No

---

## [Decision Letter · Decision Letter 1]

2 Jan 2023

PONE-D-22-02451R1Women’s empowerment and contraceptive use: recent evidence from ASEAN countriesPLOS ONE

Dear Dr. Efendi,

Thank you for submitting your manuscript to PLOS ONE. After careful consideration, we feel that it has merit but does not fully meet PLOS ONE’s publication criteria as it currently stands. Therefore, we invite you to submit a revised version of the manuscript that addresses the points raised during the review process. Please submit your revised manuscript by Feb 16 2023 11:59PM. If you will need more time than this to complete your revisions, please reply to this message or contact the journal office at plosone@plos.org. Please include the following items when submitting your revised manuscript:A rebuttal letter that responds to each point raised by the academic editor and reviewer(s). You should upload this letter as a separate file labeled 'Response to Reviewers'.A marked-up copy of your manuscript that highlights changes made to the original version. You should upload this as a separate file labeled 'Revised Manuscript with Track Changes'.An unmarked version of your revised paper without tracked changes. You should upload this as a separate file labeled 'Manuscript'.

We look forward to receiving your revised manuscript.

Kind regards,

Thae Maung Maung, MBBS, MSc (International Health), PhD

Academic Editor

PLOS ONE

Additional Editor Comments:

Please address the specific points mentioned by the reviewers.

Reviewers' comments:

Reviewer's Responses to Questions

**Comments to the Author**

1. If the authors have adequately addressed your comments raised in a previous round of review and you feel that this manuscript is now acceptable for publication, you may indicate that here to bypass the “Comments to the Author” section, enter your conflict of interest statement in the “Confidential to Editor” section, and submit your "Accept" recommendation.

Reviewer #1: All comments have been addressed

2. Is the manuscript technically sound, and do the data support the conclusions?

Reviewer #1: Partly

3. Has the statistical analysis been performed appropriately and rigorously? 

Reviewer #1: No

4. Have the authors made all data underlying the findings in their manuscript fully available?

Reviewer #1: Yes

5. Is the manuscript presented in an intelligible fashion and written in standard English?

Reviewer #1: Yes

6. Review Comments to the Author

Reviewer #1: This revised manuscript assesses the relationship between measures of women's empowerment and contraceptive use in five ASEAN countries. The topic is important, and is understudied in the focal region. Below are comments that I hope will further improve the contribution of this manuscript.

1. Describing the countries included in the analysis. A Table with key national indicators and sample characteristics, for each included country would help to situate the analysis in regional context.

2. A clearer description of the outcome variable--contraceptive use is needed. What specific methods were asked in each country? Is the measure constructed to be comparable across settings?

3. The conceptual framework should appear in the background, and measures for empowerment should align with Kabeer's framework, as presented in the background. For example, women's labor force participation in this model arguably is a resource for empowerment; whereas, disagreement with wive beating, decision-making, and "knowledge" are meaures of individual agency. These measures might be more clearly conceptually organized? Also, whey is women's schooling not included as a resource for their empowerment in the model?

4. A much clearer description for how the empowerment variables are constructed is needed. For example, how are the 5 IPV attitudes items coded and combined. This is not described in any detail in the manuscript, and the construction of the other empowerment variables also need more detail.

5. A thorough editing of the manuscript is needed throughout for clarity of exposition.

6. The description of the sample sizes for each country should appear in the methods, with clarity about the original number of eligible women for each country and numbers of women dropped due to any missing data.

7. The description of the methods of analysis is very limited and needs more detail and clarity of writing.

8. Interpretation of the findings needs some work. For example, these are cross-sectional surveys, and so it would be more appropriate to speak of higher/lower contraceptive use by covariates of interest, rather than increases/decreases. This interpretation also aligns with the associational nature of the findings from a cross-sectional study.

9. More nuanced discussion of the similarities and differences in the findings across ASEAN settings would be helpful.

7. PLOS authors have the option to publish the peer review history of their article (what does this mean?). If published, this will include your full peer review and any attached files.

Reviewer #1: No

---

## [Author Response · Author response to Decision Letter 1]

17 Feb 2023

To: Thae Maung Maung, MBBS, MSc (International Health), PhD

Academic Editor

PLOS ONE

Re: Revised Manuscript 

Date: 23 January 2023

————————————————————————————————————

Dear Academic Editor

Thank you for your email dated 16 January 2023 with the reviewer comments for the manuscript titled “Women’s empowerment and contraceptive use: recent evidence from ASEAN countries” (Manuscript ID PONE-D-22-02451R1). As directed by your correspondence, we are submitting the revised manuscript with changes highlighted in red colour that addresses all reviewer comments and suggestions. Below, please find our responses to the reviewer’s comments. We thank you for the opportunity to submit this revised manuscript.

Comments to the Author

No. Comment Response

 Additional Editor Comments: 

• An unmarked version of your revised paper without tracked changes. You should upload this as a separate file labelled 'Manuscript'.

Thank you very much for your suggestion. We have prepared the files as suggested. 

 Reviewer's Responses to Questions 

1. If the authors have adequately addressed your comments raised in a previous round of review and you feel that this manuscript is now acceptable for publication, you may indicate that here to bypass the “Comments to the Author” section, enter your conflict of interest statement in the “Confidential to Editor” section, and submit your "Accept" recommendation.

Reviewer #1: All comments have been addressed Thank you for the comments.

2. Is the manuscript technically sound, and do the data support the conclusions?

Reviewer #1: Partly

 Thank you for the comments.

3. Has the statistical analysis been performed appropriately and rigorously? 

Reviewer #1: No Thank you for the comments. 

4. Have the authors made all data underlying the findings in their manuscript fully available?

Reviewer #1: Yes Thank you for the comments. 

5. Is the manuscript presented in an intelligible fashion and written in standard English?

Reviewer #1: Yes

 Thank you for the comments. 

6. PLOS authors have the option to publish the peer review history of their article (what does this mean?). If published, this will include your full peer review and any attached files.

Do you want your identity to be public for this peer review? For information about this choice, including consent withdrawal, please see our Privacy Policy.

Reviewer #1: No Thank you for the comments. 

 Review Comments to the Author

Reviewer #1: This revised manuscript assesses the relationship between measures of women's empowerment and contraceptive use in five ASEAN countries. The topic is important, and is understudied in the focal region. Below are comments that I hope will further improve the contribution of this manuscript: Thank you very much for the suggestions. 

1. Describing the countries included in the analysis. A Table with key national indicators and sample characteristics, for each included country would help to situate the analysis in regional context.

 Thanks for the suggestion, we have added key national indicators for each country in table 1.

2. A clearer description of the outcome variable--contraceptive use is needed. What specific methods were asked in each country? Is the measure constructed to be comparable across settings?

 Thank you, we have added the clearer description of the outcome variable—contraceptive use in line 147-153.

3. The conceptual framework should appear in the background, and measures for empowerment should align with Kabeer's framework, as presented in the background. For example, women's labor force participation in this model arguably is a resource for empowerment; whereas, disagreement with wive beating, decision-making, and "knowledge" are meaures of individual agency. These measures might be more clearly conceptually organized?

Also, whey is women's schooling not included as a resource for their empowerment in the model?

 Thank you for the suggestion, we have added the conceptual framework which align with Kabeer’s framework as described in the background section.

Women's schooling is included in the component of women's knowledge as a measure of empowerment, this explained in the method section.

4. A much clearer description for how the empowerment variables are constructed is needed. For example, how are the 5 IPV attitudes items coded and combined. This is not described in any detail in the manuscript, and the construction of the other empowerment variables also need more detail.

 Many thanks, we have added the clearer description for each empowerment variables in line 167-199.

5. A thorough editing of the manuscript is needed throughout for clarity of exposition.

 Many thanks for the comment, we have edited throughout the text

6. The description of the sample sizes for each country should appear in the methods, with clarity about the original number of eligible women for each country and numbers of women dropped due to any missing data.

 Thank you for the comment, we have added the description of the sample size with clarity of the original and actual sample size used in the study in line 124-134.

7. The description of the methods of analysis is very limited and needs more detail and clarity of writing.

 Thank you for the comment we have added more detailed of analysis methods in line 217-221

8. Interpretation of the findings needs some work. For example, these are cross-sectional surveys, and so it would be more appropriate to speak of higher/lower contraceptive use by covariates of interest, rather than increases/decreases. This interpretation also aligns with the associational nature of the findings from a cross-sectional study.

 Thank you for the suggestion, we have revised in line 231-240

9. More nuanced discussion of the similarities and differences in the findings across ASEAN settings would be helpful.

 Many thanks for the improvement comment, we have added some explanation about the unique information of each countries to enrich the discussion.

---

## [Decision Letter · Decision Letter 2]

10 May 2023

PONE-D-22-02451R2Women’s empowerment and contraceptive use: recent evidence from ASEAN countriesPLOS ONE

Dear Dr. Efendi,

Thank you for submitting your manuscript to PLOS ONE. After careful consideration, we feel that it has merit but does not fully meet PLOS ONE’s publication criteria as it currently stands. Therefore, we invite you to submit a revised version of the manuscript that addresses the points raised during the review process.

Your revised manuscript looks much improved but there are still minor editions. Please carefully address those points.==============================

We look forward to receiving your revised manuscript.

Kind regards,

Thae Maung Maung, MBBS, MSc (International Health), PhD

Academic Editor

PLOS ONE

Journal Requirements:

Reviewers' comments:

Reviewer's Responses to Questions

**Comments to the Author**

1. If the authors have adequately addressed your comments raised in a previous round of review and you feel that this manuscript is now acceptable for publication, you may indicate that here to bypass the “Comments to the Author” section, enter your conflict of interest statement in the “Confidential to Editor” section, and submit your "Accept" recommendation.

Reviewer #3: All comments have been addressed

2. Is the manuscript technically sound, and do the data support the conclusions?

Reviewer #3: Yes

3. Has the statistical analysis been performed appropriately and rigorously? 

Reviewer #3: Yes

4. Have the authors made all data underlying the findings in their manuscript fully available?

Reviewer #3: Yes

5. Is the manuscript presented in an intelligible fashion and written in standard English?

Reviewer #3: No

6. Review Comments to the Author

Reviewer #3: Authors adequately addressed previous comments.

However, the manuscript still needs thorough copy editing to remove few grammatical errors.

Please remove the subtitles in the Discussion section.

7. PLOS authors have the option to publish the peer review history of their article (what does this mean?). If published, this will include your full peer review and any attached files.

Reviewer #3: No

---

## [Author Response · Author response to Decision Letter 2]

24 May 2023

To: Thae Maung Maung, MBBS, MSc (International Health), PhD

Academic Editor

PLOS ONE

Re: Revised Manuscript 

Date: 24 May 2023

————————————————————————————————————

Dear Academic Editor

Thank you for your email dated 11 May 2023 with the reviewer comments for the manuscript titled “Women’s empowerment and contraceptive use: recent evidence from ASEAN countries” (Manuscript ID [PONE-D-22-02451R2). As directed by your correspondence, we are submitting the revised manuscript with changes highlighted in red colour that addresses all reviewer comments and suggestions. Below, please find our responses to the reviewer’s comments. We thank you for the opportunity to submit this revised manuscript.

Comments to the Author

No. Comment Response

 Additional Editor Comments: 

• An unmarked version of your revised paper without tracked changes. You should upload this as a separate file labelled 'Manuscript'.

Answer: Thank you very much for your suggestion. We have prepared the files as suggested. 

2. Journal Requirements:

Answer: Thank you for your comments on our reference list in our paper. We appreciate your thorough analysis and would want to note that we have already checked that our reference list is entire, correct, and accurate.

Reviewer's Responses to Questions 

1. If the authors have adequately addressed your comments raised in a previous round of review and you feel that this manuscript is now acceptable for publication, you may indicate that here to bypass the “Comments to the Author” section, enter your conflict of interest statement in the “Confidential to Editor” section, and submit your "Accept" recommendation.

Reviewer #3: All comments have been addressed

answer: Thank you for the feedback

2. Is the manuscript technically sound, and do the data support the conclusions?

Reviewer #3: Yes

Answer: Thank you for the feedback

3. Has the statistical analysis been performed appropriately and rigorously?

Reviewer #3: Yes

answer: Thank you for the feedback

4. Have the authors made all data underlying the findings in their manuscript fully available?

Reviewer #3: Yes 

answer: Thank you for the feedback

5. Is the manuscript presented in an intelligible fashion and written in standard English?

Reviewer #3: No 

answer: Thank you for the valuable comment, we have made a correction through our manuscript, and we have done made the revision for improving our English written.

6. Review Comments to the Author

Reviewer #3: Authors adequately addressed previous comments.

However, the manuscript still needs thorough copy editing to remove few grammatical errors.

Please remove the subtitles in the Discussion section.

answer: Thank you for the suggestion, we have removed all the subtitled in the discussion section.

7. PLOS authors have the option to publish the peer review history of their article (what does this mean?). If published, this will include your full peer review and any attached files.

Do you want your identity to be public for this peer review? For information about this choice, including consent withdrawal, please see our Privacy Policy.

Reviewer #3: No 

answer: Thank you very much for the feedback

---

## [Decision Letter · Decision Letter 3]

6 Jun 2023

Women’s empowerment and contraceptive use: recent evidence from ASEAN countries

PONE-D-22-02451R3

Dear Dr. Efendi,

We’re pleased to inform you that your manuscript has been judged scientifically suitable for publication and will be formally accepted for publication once it meets all outstanding technical requirements.

Kind regards,

Thae Maung Maung, MBBS, MSc (International Health), PhD

Academic Editor

PLOS ONE

Additional Editor Comments (optional):

Reviewers' comments:

Reviewer's Responses to Questions

**Comments to the Author**

1. If the authors have adequately addressed your comments raised in a previous round of review and you feel that this manuscript is now acceptable for publication, you may indicate that here to bypass the “Comments to the Author” section, enter your conflict of interest statement in the “Confidential to Editor” section, and submit your "Accept" recommendation.

Reviewer #3: All comments have been addressed

2. Is the manuscript technically sound, and do the data support the conclusions?

Reviewer #3: Yes

3. Has the statistical analysis been performed appropriately and rigorously? 

Reviewer #3: Yes

4. Have the authors made all data underlying the findings in their manuscript fully available?

Reviewer #3: Yes

5. Is the manuscript presented in an intelligible fashion and written in standard English?

Reviewer #3: Yes

6. Review Comments to the Author

Reviewer #3: All comments are adequately addressed. The manuscript is technically sound and the data support conclusions. The statistical analysis has been performed appropriately and rigorously. The manuscript is presented in an intelligible fashion and written in standard English.

7. PLOS authors have the option to publish the peer review history of their article (what does this mean?). If published, this will include your full peer review and any attached files.

Reviewer #3: No

---

## [Editor Report · Acceptance letter]

19 Jun 2023

PONE-D-22-02451R3 

Women’s empowerment and contraceptive use: recent evidence from ASEAN countries 

Dear Dr. Efendi:

I'm pleased to inform you that your manuscript has been deemed suitable for publication in PLOS ONE. Congratulations! Your manuscript is now with our production department. 

Kind regards, 

on behalf of

Dr. Thae Maung Maung 

Academic Editor

PLOS ONE